# AN IMPLICIT FUNCTION LEARNING APPROACH FOR PARAMETRIC MODAL REGRESSION

## ABSTRACT

For multi-valued functions—such as when the conditional distribution on targets given the inputs is multi-modal—standard regression approaches are not always desirable because they provide the conditional mean. Modal regression approaches aim to instead find the conditional mode, but are restricted to nonparametric approaches. Such approaches can be difficult to scale, and make it difficult to benefit from parametric function approximation, like neural networks, which can learn complex relationships between inputs and targets. In this work, we propose a parametric modal regression algorithm, by using the implicit function theorem to develop an objective for learning a joint parameterized function over inputs and targets. We empirically demonstrate on several synthetic problems that our method (i) can learn multi-valued functions and produce the conditional modes, (ii) scales well to high-dimensional inputs and (iii) is even more effective for certain uni-modal problems, particularly for high frequency data where the joint function over inputs and targets can better capture the complex relationship between them. We conclude by showing that our method provides small improvements on two regression datasets that have asymmetric distributions over the targets.

## 1 INTRODUCTION

The goal in regression is to find the relationship between the input (observation) variable $X \in \mathcal{X}$ and the output (response) $Y \in \mathcal{Y}$ variable, given samples of $(X, Y)$. The underlying premise is that there exists an unknown underlying function $g^* : \mathcal{X} \mapsto \mathcal{Y}$ that maps the input space $\mathcal{X}$ to the output space $\mathcal{Y}$. We only observe a noise-contaminated value of that function: sample $(x, y)$ has $y = g^*(x) + \eta$ for some noise $\eta$. If the goal is to minimize expected squared error, it is well known that $\mathbb{E}[Y|x]$ is the optimal predictor (Bishop, 2006). It is common to use Generalized Linear Models (Nelder & Wedderburn, 1972), which attempt to estimate $\mathbb{E}[Y|x]$ for different uni-modal distribution choices for $p(y|x)$, such as Gaussian ($l_2$ regression) and Poisson (Poisson regression). For multi-modal distributions, however, predicting $\mathbb{E}[Y|x]$ may not be desirable, as it may correspond to rarely observed $y$ that simply fall between two modes. Further, this predictor does not provide any useful information about the multiple modes.

Modal regression is designed for this problem, and though not widely used in the general machine learning community, has been actively studied in statistics. Most of the methods are non-parametric, and assume a single mode jae Lee (1989); Lee & Kim (1998); Kemp & Silva (2012); Yu & Aristodemou (2012); Yao & Li (2014); Lv et al. (2014); Feng et al. (2017). The basic idea is to adjust target values towards their closest empirical conditional modes, based on a kernel density estimator. These methods rely on the chosen kernel and may have issues scaling to high-dimensional data due to issues in computing similarities in high-dimensional spaces. There is some recent work using quantile regression to estimate conditional modes (Ota et al., 2018), and though promising for a parametric approach, is restricted to linear quantile regression.

A parametric approach for modal regression would enable these estimators to benefit from the advances in learning functions with neural networks. The most straightforward way to do so is to learn a mixture distribution, such as with conditional mixture models with parameters learning by a neural network (Powell, 1987; Bishop, 1994; Williams, 1996; Husmeier, 1997; Husmeier & Taylor, 1998; Zen & Senior, 2014; Ellefsen et al., 2019). The conditional modes can typically be extracted from such models. Such a strategy, however, might be trying to solve a harder problem than is strictly

needed. The actual goal is to simply identify the conditional modes, without accurately representing the full conditional distribution. Training procedures for the conditional distribution can be more complex. Methods like EM can be slow (Vlassis & Krose, 1999) and some approaches have opted to avoid this altogether by discretizing the target and learning a discrete distribution (Weigend & Srivastava, 1995; Feindt, 2004). Further, the mixture requires particular probabilistic choices to be made, including the number of components, which may not be correctly specified: they might be more or less than the true number of conditional modes.

In this paper, we propose a new parametric modal regression approach, by developing an objective to learn a parameterized function $f(x, y)$ on both input feature and target/output. We use the Implicit Function Theorem (Munkres, 1991), which states that if we know the input-output relation in the form of an implicit function, then a general multi-valued function, under certain gradient conditions, can locally be converted to a single-valued function. We learn a function $f(x, y)$ that approximates such local functions, by enforcing the gradient conditions. We empirically demonstrate that our method can effectively learning the conditional modes on several synthetic problems, and that for those same problems, scales well when the input is made high-dimensional. We also show an interesting benefit that the joint representation learned over $x$ and $y$ appears to improve prediction performance even for uni-modal problem, for high frequency functions where the function values changes quickly between nearby $x$. Finally, we show that our method provides small improvements on two regression datasets that have asymmetric distributions over the targets. The proposed approach to multi-valued prediction is flexible, allowing for a variable number of conditional modes to be discovered for each $x$, and we believe it is a promising direction for further improvements in parametric modal regression.

## 2 PROBLEM SETTING

We consider a standard learning setting where we observe a dataset of $n$ samples, $\mathcal{S} = \{(x_i, y_i)\}_{i=1}^n$. Instead of the standard regression problem, however, we tackle the modal regression problem. The goal in modal regression is to find the set of conditional modes

$$M(x) = \left\{ y : \frac{\partial p(x, y)}{\partial y} = 0, \ \frac{\partial^2 p(x, y)}{\partial y^2} < 0 \right\} \tag{1}$$

$M(x)$ is in general a multi-valued function. Consider the example in Figure 1. There are two conditional modes for a given $x$. For example, for $x = 0$, the two conditional modes are $y_1 = -1.0$ and $y_2 = 1.0$.

The standard approaches to find these conditional modes involve learning $p(y|x)$ or to use non-parametric methods directly estimate these conditional modes. For example, for a conditional Gaussian Mixture Model, a relatively effective approximation of these modes are the means of the conditional means. More generally, to get precise estimates, non-parametric algorithms are used, like the mean-shift algorithm (Yizong Cheng, 1995). We refer readers to Chen (2018); Chen et al. (2014) for a detailed nice review. These algorithms attempt to cluster points based on $x$ and $y$, to find these conditional modes.

Looking at the plot in Figure 1, however, a natural idea is to instead directly learn a parameterized function $f(x, y)$ that captures the relationship between $x$ and $y$. Unfortunately, it is not obvious how to do so, nor how to use $f(x, y)$ to obtain the conditional modes. In the next section, we develop an approach to learn a parameterized $f(x, y)$ that can be used to extract conditional modes, by using the implicit function theorem.

## 3 AN IMPLICIT FUNCTION LEARNING APPROACH

In this section, we develop an objective to facilitate learning parametric functions for modal regression (for multi-valued prediction). The idea is to directly learn a parameterized function $f(x, y)$, where the set of minimum $y$ for a function of $f(x, y)$ corresponds to the conditional modes. The approach allows for a variable number of conditional modes for each $x$. Further, it allows us to take advantage of general parametric function approximators, like neural networks, to identify these modal manifolds that capture a smooth relationship between the conditional modes and $x$.

Consider learning an $f(x, y)$ such that $f(x, y) = 0$ for all conditional modes and non-zero otherwise. For example, for the Circle problem, $f(x, y) = x^2 + y^2 - 1$ for all conditional modes $y$. Such a

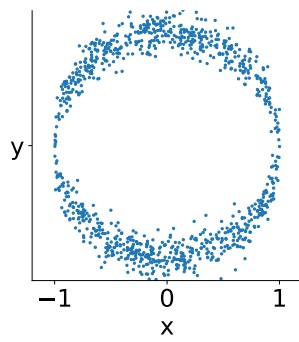

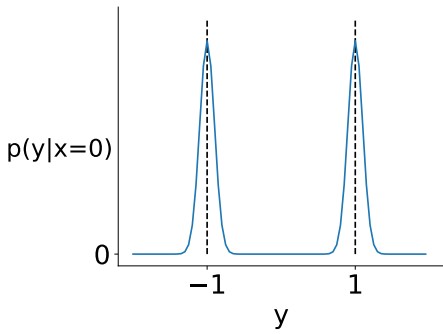

(a) A dataset looks like a circle

(b) Conditional distribution $p(y|x=0)$

Figure 1: (a) shows a dataset generated by uniformly sampling $x \in (-1, 1)$, conditioned which $y$ is sampled from $0.5N(\sqrt{1-x^2}, 0.1^2) + 0.5N(-\sqrt{1-x^2}, 0.1^2)$. (b) shows the conditional distribution when $x = 0$.

strategy—finding $f(x, y) = 0$ for all conditional modes—is flexible in that it allows for a different number of conditional modes for each $x$. The difficulty with learning such an $f$, particularly under noisy data, is constraining it to be zero for conditional modes $y_j$ and non-zero otherwise.

To obtain meaningful conditional modes $y_1, \ldots, y_{m_x}$ for $x$, the $y$ around each $y_j$ should be described by the same mapping $g_j(x)$. The existence of such $g_j$ is guaranteed by the Implicit Function Theorem (Munkres, 1991), under one condition on $f$.

**Implicit Function Theorem:** Let $f : \mathbb{R}^d \times \mathbb{R}^k \mapsto \mathbb{R}^k$ be a continuously differentiable function. Fix a point $(x, y) \in \mathbb{R}^d \times \mathbb{R}^k$ such that $f(x, y) = \mathbf{0}$, for $\mathbf{0} \in \mathbb{R}^k$. If the Jacobian matrix $J$, where the element in the $i$th row and $j$th column is $J_{[ij]} = \frac{\partial f(x,y)[i]}{\partial y[j]}$, is invertible, then there exists open sets $\mathcal{U}, \mathcal{V}$ containing $(x, y)$ such that there exists a unique continuously differentiable function $g : \mathcal{U} \mapsto \mathcal{V}$ satisfying $g(x) = y$ and $F(x, g(x)) = \mathbf{0}$.

The theorem states that if we know the relationship between independent variable $x$ and dependent variable $y$ in the form of implicit function $f(x, y) = 0$, then under certain conditions, we can guarantee the existence of some function defined locally to express $y$ given $x$. For example, a circle on two dimensional plane can be expressed as $\{(x, y)|x^2 + y^2 = 1\}$, but there is no definite expression (single-valued function) for $y$ in terms of $x$. However, given a specific point on the circle $(x_0, y_0)$ (other than $y_0$), there exists an explicit function defined locally around $(x_0, y_0)$ to express $y$ in terms of $x$. Notice that at $y_0 = 0$, the condition required by the implicit function theorem is not satisfied: $\frac{\partial(x^2+y^2-1)}{\partial y} = 2y = 0$ at $y_0 = 0$, and so is not invertible.

Obtaining such smooth local functions $g$ enables us to find these smooth modal manifolds. The conditional modes $g_1, \ldots, g_{m_x}$ satisfy $f(x, g_j(x)) = 0$ and $\frac{\partial f(x,g_j(x))}{\partial y} \neq 0$. When training $f$, we can attempt to satisfy both conditions to ensure existence of the $g_j$. The gradient condition ensures that for $y$ locally around $f(x, g_j(x))$, we have $f(x, y) \neq 0$. This encourages the other requirement that $f(x, y)$ be non-zero for the $y$ that are not conditional modes. Notice, though, that this condition is only local and may not encourage a minimal set of conditional modes. We can also add a negative sampling component that encourages all $y$ to by default have $f(x, y)^2 > 0$. We find empirically, however, that this addition is not necessary. We therefore pursue this simpler objective, which avoids the complications of negative sampling.

Now we derive the full objective, under stochastic targets. To do so, we make some assumptions on the noise around the conditional modes. In particular, we assume that the noise around each conditional mode is Gaussian. More precisely, define

$$\epsilon(X, Y) \stackrel{\text{def}}{=} g_j(X) - Y \qquad (2)$$

for $g_j$ the conditional mode for $Y$. Our goal is to approximate $\epsilon(x, y)$ with parameterized function $f_\theta(x, y)$ for parameters $\theta$. We assume

$$\epsilon(X, Y) \sim \mathcal{N}(\mu = \mathbf{0}, \sigma^2 \mathbf{I}) = (2\pi\sigma)^{-k/2} \exp\left(-\frac{\epsilon(X, Y)^2}{2\sigma^2}\right) \tag{3}$$

We also assume that $\frac{\partial \epsilon(x,y)[i]}{\partial y[j]} = 0$ for $i \neq j$: the conditional mode for dimension $i$ is not influenced by changes to the $y$ in dimension $j$. Consequently, the implicit function condition becomes $\frac{\partial \epsilon(x,y[j])}{\partial y[j]} \neq 0$, to ensure invertibility. In fact, for this function, we have

$$\frac{\partial \epsilon(x, y)}{\partial y} = \frac{\partial g_j(x) - y}{\partial y} = -1. \tag{4}$$

Therefore, when learning $f$, we simply need to constrain $\frac{\partial f_\theta(x,y)}{\partial y} = -1$ for all $y$. Putting this together, our goal is to minimize the negative log likelihood of $f_\theta$, which approximates a zero-mean Gaussian random variable, under this constraint which we encourage with a quadratic penalty term. This gives the following objective, where the goal is to find $\arg\min_\theta L(\theta)$:

$$L(\theta) \stackrel{\text{def}}{=} \sum_{i=1}^{n} f_\theta(x_i, y_i)^2 + \left\| \frac{\partial f_\theta(x_i, y_i)}{\partial y} + 1 \right\|_2^2 \tag{5}$$

The same objective is used when doing prediction, but now optimizing over $y$. Given $x^*$, we compute a $y^*$ in the set $\arg\min_y f_\theta(x^*, y)^2 + (\frac{\partial f_\theta(x^*,y)}{\partial y} + 1)^2$. These $y^*$ should correspond to the conditional modes, because the objective should be minimal for conditional modes. One option is to use gradient descent to find these $y^*$. In all our experiments, we opted for the simple strategy of searching over 200 evenly spaced values in the range of $y$.

## 4 THE PROPERTIES OF IMPLICIT FUNCTION LEARNING

In this section, we conduct experiments to investigate the properties of our learning objective. First, by using the Circle datasets, we show its utility for dealing with multimodal distribution, particularly compared to modeling the entire distribution with mixture distributions. Second, inspired by the first experiment, we show that our algorithm achieves superior performance when the underlying true function is a high frequency function. We empirically show that this is because our algorithm can better leverage the NN's representation power than the $l_2$ regression. Hence, even with regular single-modal datasets, our algorithm could still be beneficial in some cases.

### 4.1 THE CIRCLE DATASETS

The purpose of this experiment is to study the properties of our algorithm for modal prediction problems. Mixture Density Networks (**MDN**) (Bishop, 1994) are used as a baseline, to compare to an approach that learns the distribution. For both algorithms, we use two hidden layer neural network ($16 \times 16$ tanh units) and train by 128-mini-batch-size stochastic gradient descent. We optimize both algorithms by sweeping learning rate from $\{0.1, 0.01, 0.001, 0.0001\}$. For the purpose of evaluating prediction performance by testing error, we compute the root mean squared error (RMSE) for the predicted value and the true value, and the true value is defined as the one closer to the predicted value. For **MDN** to predict mode, given a point $x$, we search over 200 evenly spaced $y$ values to maximize the learned log likelihood. We refer readers to the Appendix B for a study of the sensitivity of parameter in **MDN**.

We conduct two experiments on a single-circle (two modals) and a double-circle (four modals at most) datasets respectively. On the latter, we further conduct a challenging experiment by projecting the one dimensional feature value to 128 dimensional binary feature through tile coding[1]. We empirically show that: 1) our algorithm (**Implicit**) achieves higher sample efficiency than MDN; 2) on both datasets, our algorithm uses the same parameter settings, while MDN is quite sensitive to the number of mixture components and has a larger variance across different random seeds; 3) our algorithm can scale to high dimensional feature space and still maintain good performance.

---

[1]We refer readers to http://www.incompleteideas.net/tiles.html for more details about tile coding. It is a frequently used feature generation method in reinforcement learning.

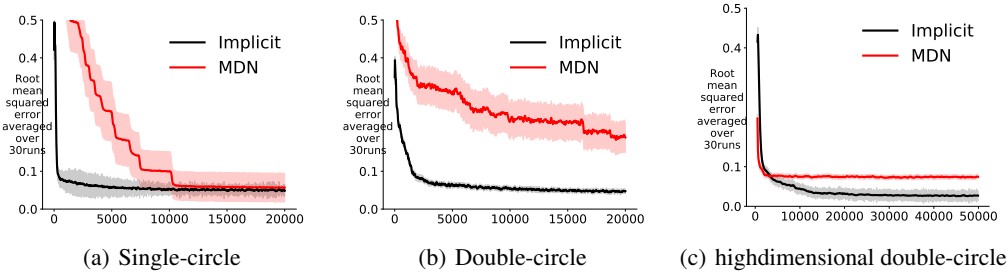

(a) Single-circle        (b) Double-circle        (c) highdimensional double-circle

Figure 2: (a)(b) shows learning curves on single-circle and double-circle datasets respectively. In the single-circle problem, **MDN** uses 3 mixture components for learning; on the double-circle dataset and the high dimensional variant, it uses 4 components to learn. All results are averaged over 30 random seeds and the shaded area indicates standard error.

**Single-circle.** The training set is acquired by uniformly sampling $40,000$ data points from a circle: $\{(x,y)|x^2 + y^2 = 1\}$. Note that, since we add zero mean Gaussian noise with standard deviation $\sigma = 0.1$ to targets, we can interpret the conditional probability distribution $p(y|x)$ as a two-component mixture Gaussian $p(y|x) = 0.5N(y; \sqrt{1-x^2}, \sigma^2) + 0.5N(y; -\sqrt{1-x^2}, \sigma^2)$ as shown in Figure 1. On this dataset, though there are only two modals, **MDN** does badly (i.e. the testing error is almost outside of figure) with only two mixture components, hence we use three components for it. The learning curves are shown in Figure 2. One can see that our algorithm learns much faster and is more stable in term of the standard deviation across different random seeds. Figure 3(a)(b) show the predictions outputted by our approach and **MDN** at the end of training (i.e. after 20000 updates).

**Double-circle.** On the double-circle dataset, the same number of training points are randomly sampled from two circles (i.e. $\{(x,y)|x^2 + y^2 = 1\}, \{(x,y)|x^2 + y^2 = 4\}$) and with the same Gaussian noise added to targets. This should be a challenging dataset where $p(y|x)$ can be considered as a *piece-wise* mixture of Gaussian: there are four components on $x \in (-1, 1)$ and two components on $x \in (-2, -1) \cup (1, 2)$. As a result, we set number of components as $4$ for our competitor **MDN** and keep all the parameter setting the same for our algorithm. It should be noted that, the algorithm **MDN** learns much slower when moving from the two-components single-circle dataset to the four-components double-circle datasets. Additionally, we find **MDN** requires more number of components than the true distribution actually has to achieve superior performance. Similar to the single-circle case, Figure 3(c)(d) show the predictions at the end of training, from which we can see that the our algorithm does significantly better than the competitor.

**High dimensional double circle.** After using tile coding, we project the original one dimensional $x \in [-2, 2]$ to binary features $\phi(x) \in \{0, 1\}^{128}$. Then the projected features are used as input to the neural networks. One can see that although both algorithms converge faster, **MDN** converges to a worse solution than our algorithm does. To our best knowledge, this is the first time modal regression is tested on such high dimensional dataset; most modal regression algorithms have been tested on low-dimensional problems.

**Verify the learned error distribution.** On both datasets, we investigate the validity of the assumption on the error distribution $\epsilon(X, Y)$, by examining the empirical density of our learned error function $f_\theta(x, y)$ using all examples from training set. Figure 4 shows the error distribution on the training set without noise (a) and with noise (b) added to the target variable. It is obvious that the error function indeed looks Gaussian distributed and the one trained without noise (4(a)) shows an extremely small variance.

## 4.2 ROBUSTNESS TO HIGH FREQUENCY DATA

The above circle example can be thought of as an extreme case where the underlying true function has extremely high, or even unbounded frequency (i.e. when the input changes a little, there is a

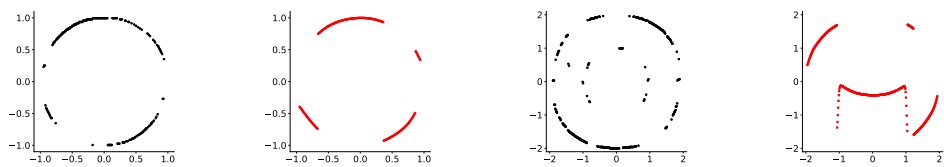

(a) **Implicit**, single circle  (b) **MDN**, single circle  (c) **Implicit**, double circle  (d) **MDN**, double circle

Figure 3:  (a)(b) shows the predictions of both our algorithm and MDN on the single circle dataset. (c)(d) show the predictions of the two algorithms on the double circle dataset.

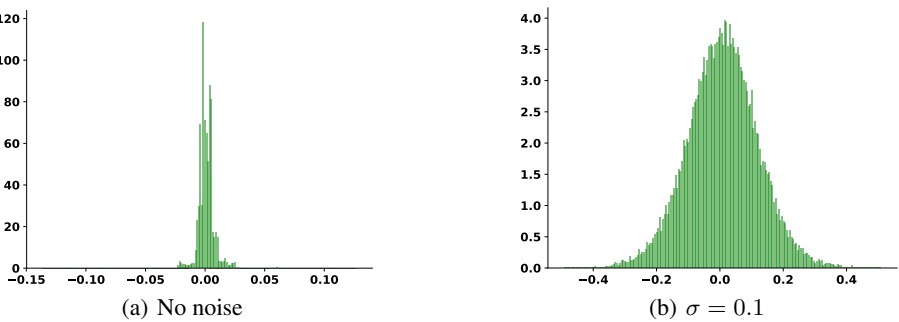

(a) No noise

(b) $\sigma = 0.1$

Figure 4:  (a)(b) shows the empirical density of $f_\theta(x, y)$ for $(x, y)$s in the training set $\mathcal{S}$ when the algorithm is trained by not adding noise to target and adding noise (standard deviation $\sigma = 0.1$) respectively.

sharp change for the true target). As a result, in this section, we provide a less extreme dataset and show that our algorithm has advantage dealing with such problems.

We generate a synthetic dataset by uniformly sampling $x \in [-2.5, 2.5]$ and using the below underlying procedure to compute the targets:

$$y = \begin{cases} \sin(8\pi x) + \xi & x \in [-2.5, 0) \\ \sin(0.5\pi x) + \xi & x \in [0, 2.5] \end{cases} \tag{6}$$

where $\xi$ is zero-mean Gaussian noise with variance $\sigma^2$. This function has relatively high frequency when $x \in [-2.5, 0)$ and has a relatively low frequency when $x \in [0, 2.5]$.

**Significance of this artificial dataset.**   The dataset is designed to be difficult to learn, because several existing works (Smale et al., 2004; Smale & Zhou, 2005; Jiang, 2019) indicate that the bandwidth limit of the underlying true function strongly affects the sample efficiency of a learning algorithm. Intuition about why high frequency functions (large bandwidth limit) are difficult to learn can be gained from the Shannon sampling theorem, which characterizes the relation between minimum sampling rate[2] to perfectly recover a signal and the bandwidth limit of the signal (Zayed, 1993): the sampling rate should exceed twice of the maximum frequency of the signal to guarantee perfect signal reconstruction.

**Examining the learning behaviour.**   We use $16 \times 16$ hidden tanh units NN for our algorithm. For the $l_2$ regression, we use the same size NN and perform extensive parameter sweep to optimize its performance: activation function type swept over tanh and relu, learning rate swept over $\{0.1, 0.01, 0.001, 0.0001, 0.00001\}$. For both algorithms, we used a mini-batch size of 128. Note the only difference between the two NNs is that **Implicit** has one more input unit, i.e. the target $y$.

---

[2]In signal processing, sampling refers to the reduction of a continuous-time signal to discrete time signal. Sampling rate refers to number of samples per second.

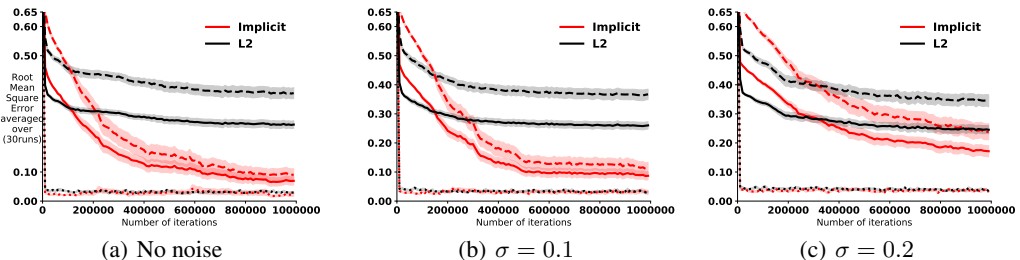

(a) No noise          (b) $\sigma = 0.1$          (c) $\sigma = 0.2$

Figure 5: Figure (a)(b)(c) show performances of **Implicit (red)** and $l_2$ regression **L2 (black)** objective as we increase the Gaussian noise variance. We show the testing error measured by RMSE on entire testing set (**solid line**), on high frequency region (i.e. $x \in [-2.5, 0.0)$, **dashed line**) and on low frequency region ($x \in [0.0, 2.5]$, **dotted line**). The results are averaged over 30 random seeds.

Figure 5(a-c) shows the evaluation curve on testing set for the above two algorithms as the noise variance increases. The learning curve is plotted by evaluating the testing error every 10k number of iterations (i.e. mini-batch updates) averaged over 30 random seeds. We show the testing error on the entire testing set, on high frequency area ($x \leq 0$) and low frequency area ($x \geq 0$) respectively. We run into 1 million iterations to make sure each algorithm is sufficiently trained and both early and late learning behaviour can be examined.

Notice that, trained without observation noise (i.e. $\xi \equiv 0$), our implicit function learning approach achieves a much lower error (at the order of $10^{-2}$) than the $l_2$ regression does (at the order of $10^{-1}$). As noise increases, the targets likely become less informative and hence our algorithm's performance decreases to be closer to the $l_2$ regression. Unsurprisingly, for both algorithms, the high frequency area is much more difficult to learn and is a dominant source of the testing error. After sufficient training, our algorithm can finally reduce the error of both the high and low frequency regions to a similar level.

**Examining the neural network representation.** We further investigate the performance gain of our algorithm by examining the learned NN representation. We plot the predictions in figure 6(a) and the corresponding learned NN representation through heatmap in figure 6(b). In a trained NN, we consider the output of the second hidden layer as the learned representation, and investigate its property by computing pairwise distances measured by $l_2$ norm between 161 different evenly spaced points on the domain $x \in [-2.5, 2.5]$. That is, a point $(x, x')$ on the heatmap in figure 6(b) denotes the corresponding distance measured by $l_2$ norm between the NN representations of the two points (hence the heatmap shows symmetric pattern w.r.t. the diagonal).

The representations provide some insight into why Implicit outperformed l2 regression. In figure 6(a), the $l_2$ regression fails to learn one part of the space around the interval $[-2.25, -1.1]$. This corresponds to the black area in the left heatmap, implying that the $l_2$ distance between NN representations among those points are almost zero. Additionally, one can see that the heatmap of our approach shows a clearly high resolution on the high frequency area and a low resolution on the low frequency area, which coincides with our intuition for a good representation: in the high frequency region, the target value would change a lot when $x$ changes a little, so we expect those points to have finer representations than those in low frequency region. This experiment shows that given the same NN size, our algorithm is better able to leverage the representation power of the NN.

## 5 RESULTS ON STANDARD REGRESSION DATASETS

In this section, we show that our approach can still be effective even for two standard real world datasets, with comparable performance to standard regression approaches. Our appendix B.1 includes learning curves and all details for reproducing the experiments. We compare to $l_2$ and Poisson regression. For our algorithm, we use $64 \times 64$ tanh units NN. For the $l_2$ regression, we use the same size NN but we consider hidden unit types as meta-parameter and we optimize them over tanh and relu. We report both root mean squared error (RMSE) and mean absolute error (MAE) on training

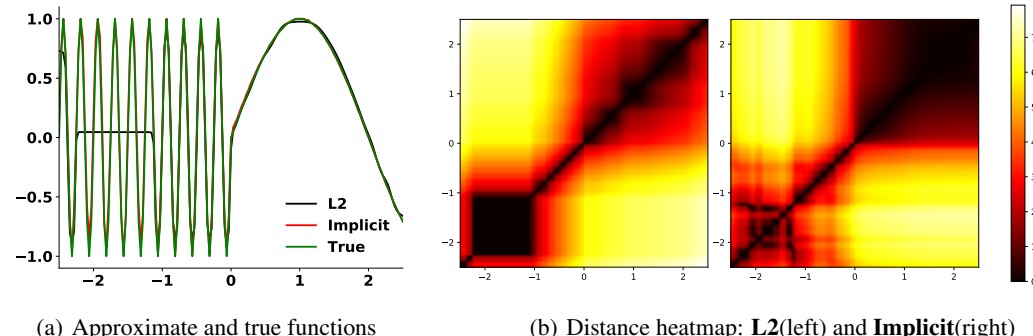

(a) Approximate and true functions         (b) Distance heatmap: **L2**(left) and **Implicit**(right)

Figure 6: (a) Approximated functions and true function. (b) The distance matrix showed in heat map computed by hidden layer representation learned by **L2** (left) and **Implicit** (right) method.

Table 1: Prediction errors on bike sharing dataset. All numbers are multiplied by $10^2$.

| Algorithms | Train RMSE | Train MAE | Test RMSE | Test MAE |
|---|---|---|---|---|
| LinearReg | 10094.40($\pm$13.60) | 7517.64($\pm$19.95) | 10129.40($\pm$59.26) | 7504.22($\pm$ 44.20) |
| LinearPoisson | 8798.26($\pm$14.58) | 5920.99($\pm$13.66) | 8864.90($\pm$66.07) | 5935.00($\pm$ 38.32) |
| NNPoisson | 1620.46($\pm$47.71) | 1071.39($\pm$29.55) | 4150.03($\pm$77.76) | 2616.49($\pm$ 20.45) |
| L2 | 708.12($\pm$28.79) | 550.14($\pm$23.64) | 3854.10($\pm$39.33) | 2560.83($\pm$ 18.30) |
| Implicit | 880.90($\pm$30.53) | 691.10($\pm$23.99) | **3683.76**($\pm$57.12) | **2426.52**($\pm$ 23.81) |

and testing set respectively. All of our results are averaged over 5 runs and at each run, the data is randomly split into training and testing sets. Algorithms and datasets are as follows.

**LinearReg.** Linear regression, i.e., ordinary least square objective. The prediction is linear in term of input features. We use this algorithm as a reference.

**LinearPoisson.** The mean of Poisson is parameterized by a linear function in term of input feature.

**NNPoisson.** The mean of Poisson is parameterized by a neural network (NN) (Fallah et al., 2009).

**Bike sharing dataset.** We use the bike sharing dataset Fanaee-T & Gama (2013) where the target is Poisson distributed to show our algorithm's generality. Our prediction task is to predict count of rental bikes in a specific hour, given the information about weather, temperature, date, etc. We show the training result in table 1 with standard errors. One can see that in the case of using linear function approximator, **LinearPoisson** has clear advantage over **LinearReg**; however, in deep learning setting, we found **NNPoisson** achieves a worse performance than the regular $l_2$ regression. Our algorithm, which does not make any assumptions on the conditional distribution $p(y|x)$, achieves slightly better performance.

**Song year dataset.** The song year dataset (Bertin-Mahieux et al., 2011) contains about half-million instances. The task is to predict a song's release year by using audio features of the song. The dataset has a target distribution for which it is not obvious which generalized linear model we should use. Hence we treat it as a regular regression dataset. One can see from Table 2 that our algorithm can slightly outperform $l_2$ regression, again potentially because we are not making distributional assumptions on $p(y|x)$.

Table 2: Prediction errors on song year dataset. All numbers are multiplied by $10^2$.

| Algorithms | Train RMSE | Train MAE | Test RMSE | Test MAE |
|---|---|---|---|---|
| LinearReg | 956.40($\pm$0.37) | 681.56($\pm$0.68) | 957.56($\pm$1.49) | 681.66($\pm$1.52) |
| L2 | 798.61($\pm$1.44) | 563.51($\pm$0.65) | 879.48($\pm$1.74) | 606.57($\pm$1.95) |
| Implicit | 822.53($\pm$1.59) | 582.11($\pm$4.18) | **869.62**($\pm$1.33) | **600.73**($\pm$2.15) |

## 6 CONCLUSION AND DISCUSSION

The paper introduces a simple and powerful implicit function learning approach for modal regression. We show that it can handle datasets where the conditional distribution $p(y|x)$ is multimodal, and is particularly useful when the underlying true mapping has a large bandwidth limit. We also illustrate that our algorithm achieves competitive performance on large real world datasets with different underlying target distributions. We would like to conclude with the following future directions.

First, it would be interesting to establish connections to KDE-based modal regression methods, which have a nice theoretical interpretation (Feng et al., 2017). The connection may yield finite sample analysis for our implicit function learning algorithm. Second, like many supervised learning algorithms, our algorithm may also overfit to noise. Popular regularization technique such as random dropout (Srivastava et al., 2014) may be tested for very noisy data. Third, in online learning setting, the efficiency of doing prediction by $\arg\min_y f_\theta(x, y)^2 + (\frac{\partial f_\theta(x,y)}{\partial y} + 1)^2$ becomes a concern. One possible solution is to borrow ideas from cross-entropy method as used in reinforcement learning (Lim et al., 2018; Simmons-Edler et al., 2019). For example, we can use a separate NN to suggest a set of initial values of $y$ for searching optimums by gradient methods. Last, it is worth investigating alternative constraints on the Jacobian instead of restricting the diagonal values to $-1$.

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

## A  APPENDIX

The appendix includes additional experimental results, and all experimental details for reproducible research.

## B  ADDITIONAL EXPERIMENTAL RESULTS

**A classic inverse problem.**  One important type of applications of multi-value function prediction is inverse problem. We now show additional results on a classical inverse function domain as used in (Bishop, 1994). The learning dataset is composed as following.

$$x = y + 0.3\sin(2\pi y) + \xi, y \in [0, 1] \tag{7}$$

where $\xi$ is a random variable representing noise with uniform distribution $U(-0.1, 0.1)$. We generate 80k training examples. In Figure 7, we plot the training dataset, and predictions by our implicit function learning algorithm with $(\arg\min_y f_\theta(x, y)^2 + (\frac{\partial f_\theta(x,y)}{\partial y} + 1)^2)$. We search over 200 evenly spaced $y$s in $[0, 1]$ for 200 evenly spaced $x \in [0, 1]$ to get points in the form of $(x, y)$s.

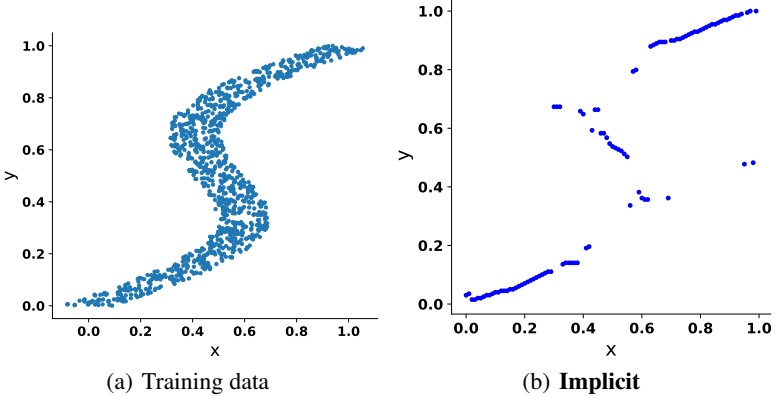

(a) Training data          (b) **Implicit**

Figure 7:  Figure (a) shows what the training data looks like. (b) shows the predictions of our implicit learning approach.

**Additional result of mixture density network.**  We now provide additional result of mixture density network **MND** on the three multivalue prediction dataset:circle, double circle, and high dimensional double circle. The learning curves are shown in Figure 8. For each component on each training dataset, we choose stepsize from $\{0.1, 0.01, 0.001, 0.0001\}$.

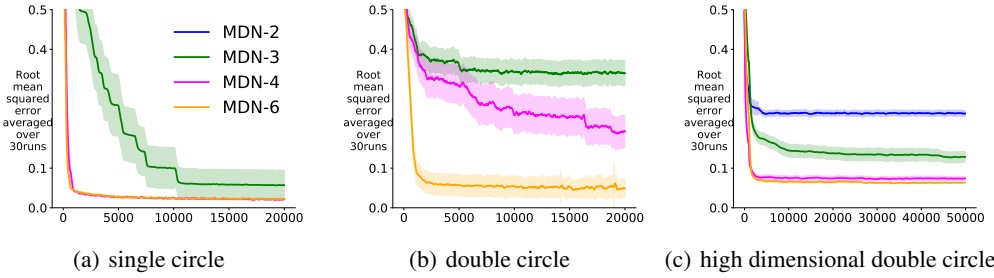

(a) single circle          (b) double circle          (c) high dimensional double circle

Figure 8:  We show the learning curves of mixture density network with different number of components. In (a) and (b), the learning curves of component 2 are out of figure. For each number of component, we choose the learning curve by optimizing other parameter (i.e. stepsize). The results are averaged over 30 random seeds.

## B.1    Reproduce experiments in the paper

In this section, we provide additional information about datasets we used and experimental details for reproducing all results in this paper. Our implementation is based on Python 3.3.6. Our deep learning implementation is based on Tensorflow 1.11.0 Abadi et al. (2015). All of our algorithms are trained by Adam optimizer (Kingma & Ba, 2015) with mini-batch size 128 and all neural networks are initialized by Xavier (Glorot & Bengio, 2010). For our implicit function learning algorithm, we use tanh units for all nodes in neural network[3] We search over 200 evenly spaced values for prediction except on song year dataset where we use 100. Best parameter settings used to reproduce experiment on each dataset are showed in figure 9. The best parameters are chosen according to the testing error at the end of learning.

| Algorithms & datasets | Bike sharing | Song year | frequency test, eq(6) | circle/double circle | inverse problem |
|---|---|---|---|---|---|
| L2 regression | learning rate = 0.0001, 64-by-64 tanh units | learning rate = 0.0001, 64-by-64 tanh units | learning rate=0.01, 16-by-16 tanh units NN | - | - |
| Implicit | learning rate = 0.0001, 64-by-64 tanh units | learning rate = 0.0001, 64-by-64 tanh units | learning rate = 0.001, 16-by-16 tanh units NN | learning rate = 0.01, 16-by-16 tanh units NN | learning rate = 0.01, 16-by-16 tanh units NN |
| LinearReg | learning rate = 0.0001 | learning rate = 0.0001 | - | - | - |
| PoissonReg | learning rate = 0.01 | - | - | - | - |
| NNPoisson | learning rate = 0.0001, 64-by-64 tanh units, linear output unit | - | - | - | - |

Figure 9: Best parameter setting for reproducing experiments.

### B.1.1    Circle and double circle experiment

Circle dataset is generated by uniformly sampling $x \in [-1, 1]$ first and then $y = \sqrt{1 - x^2}$ or $y = -\sqrt{1 - x^2}$ with equal probability. Double circle dataset is generated by uniformly sampling an angle $\alpha \in [0, 2\pi]$ then use polar expression to compute $x = r \cos \alpha, y = r \sin \alpha$ where $r = 1.0$ or $r = 2.0$ with equal probability. High dimensional double dataset is generated by mapping the original $x$ to $\{0, 1\}^{128}$ dimensional space. We refer to http://www.incompleteideas.net/tiles.html for tile coding software. The setting of tile coding we used to generate feature is: memory size $= 128$, $8$ tiles and $4$ tilings. We keep the neural network size the same as we used in low dimensional case, i.e. $16 \times 16$ tanh units.

For mixture density network (**MDN**), we use tanh hidden layers and three mixture components on single circle examples, four mixture components on double-circle example. We sweep over learning rate from $\{0.1, 0.01, 0.001, 0.0001\}$ and the best it chooses is $0.001$ on the single circle dataset and $0.01$ on the double circle dataset. The maximization is done by using MLE, the method described in the original paper Bishop (1994).

### B.1.2    High frequency data experiment

The dataset is generated by uniformly sampling $x \in [-2.5, 2.5]$ and then compute targets according to the equation 6:

$$y = \begin{cases} \sin(8\pi x) & x \in [-2.5, 0) \\ \sin(0.5\pi x) & x \in [0, 2.5] \end{cases}$$

---

[3]Rigorously, to satisfy the assumption that $f_\theta(X, Y)$ is Gaussian distributed, linear output unit should be used. However, we observe large error rarely happens. In fact, in our case, it is easy to see that assuming the distribution to be truncated Gaussian (then using tanh is justified) would yield the same optimization objective.

Table 3: Prediction errors on song year dataset with author suggested train-test split. All numbers are multiplied by $10^2$. The randomness comes from neural network initialization and stochastic mini-batch update.

| Algorithms | Train RMSE | Train MAE | Test RMSE | Test MAE |
|---|---|---|---|---|
| LinearReg | 956.43($\pm$0.05) | 680.66($\pm$0.31) | 952.22($\pm$0.09) | 681.22($\pm$0.39) |
| L2 | 836.77($\pm$0.47) | 584.00($\pm$1.22) | 888.58($\pm$0.65) | 612.79($\pm$0.67) |
| Implicit | 857.48($\pm$0.51) | 593.04($\pm$1.11) | 886.03($\pm$0.63) | 610.52($\pm$1.10) |

We sweep over $\{0.1, 0.01, 0.001, 0.0001, 0.00001\}$ to optimize stepsize for both the $l_2$ regression and our algorithm, while we additionally sweep over hidden unit and output unit type for the $l_2$ regression from {tanh, relu}. The best parameter is chosen according to the testing error at the end of training, and the testing error is averaged over 30 runs and at each run, the data is randomly split into training and testing sets. Since the best learning rate chosen chosen by the $l_2$ regression is $0.01$ while our algorithm chooses $0.001$, in figure 10, we also plot the learning curve with learning rate $0.001$ to make sure that the performance difference is not due to a slower learning rate of our algorithm.

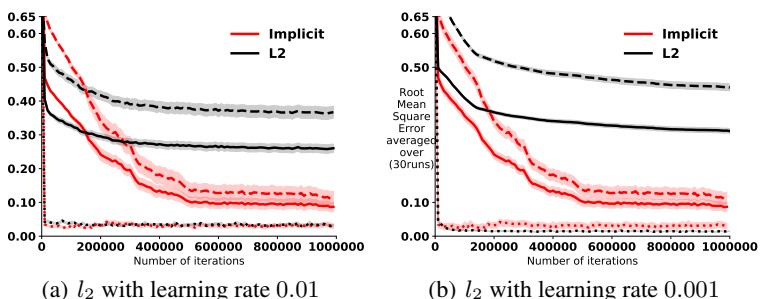

(a) $l_2$ with learning rate $0.01$        (b) $l_2$ with learning rate $0.001$

Figure 10: In (a) we repeat the figure shown in previous

### B.1.3 EXPERIMENTS ON REAL WORLD DATASETS

The bike sharing dataset (Fanaee-T & Gama, 2013) (https://archive.ics.uci.edu/ml/datasets/bike+sharing+dataset) and song year dataset Bertin-Mahieux et al. (2011) (https://archive.ics.uci.edu/ml/datasets/yearpredictionmsd) information are presented in figure 11. Note that the two datasets have very different target distributions as shown in figure 12. On the two dataset, we use $64 \times 64$ tanh hidden units and sweep over learning rate from $\{0.01, 0.001, 0.0001\}$. For the $l_2$ regression, we found that using tanh unit works better than relu.

Notice that the contributor of song year dataset suggests using the last $51630$ as testing set (Bertin-Mahieux et al., 2011), hence we also report this additional result in table 3. The performance is actually quite similar to that by random split.

Dataset and preprocessing information

| | Number of instances | Train size | Test size | Input feature dimension after preprocessing | Input feature preprocess | Target preprocess |
|---|---|---|---|---|---|---|
| **Bike sharing** | 17379 | 13903 | 3476 | 114 | remove attributes: date, index, year, weather situation 4 and weekday 7; registered, casual; use one-hot encoding for all categorical variables | Scale to [0, 1] except for poisson regression algorithms; scaled back when compute test error |
| **Song Year** | 515345 | 412276 or 463715 | 103069 or 51630 | 90 | standardize to zero-mean unit variance; statistics acquired by using training set | Scale to [0, 1]; scaled back when compute test error |

Figure 11: Data preprocessing information.

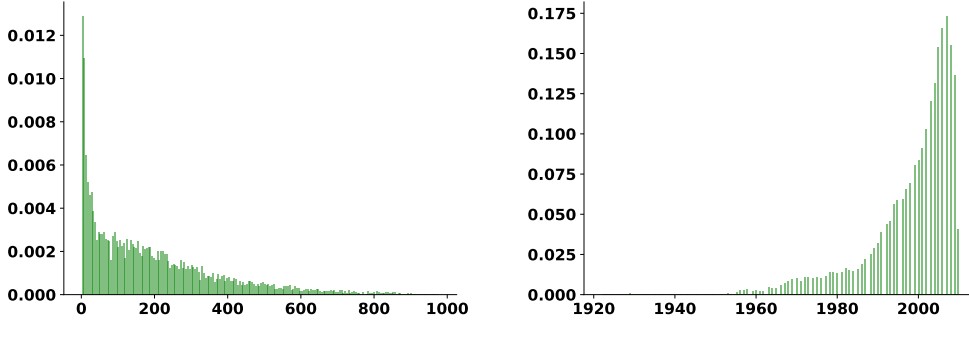

(a) Bike sharing dataset target distribution    (b) Song year dataset target distribution

Figure 12: Bike sharing targets show a clear Poisson distribution while song year dataset's target distribution is not intuitive.

