# OpenReview forum: "An implicit function learning approach for parametric modal regression"
_ICLR.cc/2020/Conference — Reject_

### Official Review · AnonReviewer2 · 2019-10-16
**Official Blind Review #2**

**Rating:** 1

**Review:**

The paper proposes an implicit function approach to learning the modes of multimodal regression. The basic idea is interesting, and is clearly related to density estimation, which the paper should have discussed. On the other hand the tackled problem is a weaker variant of the learning multimodal output distributions in regression. What is the benefit of tackling this simpler problem, when in reality we are interested in the output distributions? In Bayesian literature (BNNs, deep GPs, Bayesian regression) multimodal outputs are routinely produced, but the paper ignores these methods. The paper should compare to these, and demonstrate the benefit of just learning the modes instead of learning the full output distributions (perhaps the mode learning gives more accurate mode assignment?).

The paper should be more rigorous on what is the problem setting. Right now its unclear if g(.) is multimodal, if \eta is multimodal, or if both are. This is an important distinction since it deals with whether the underlying system is modal or if the noise modal, and connects with aleatoric/epistemic characterisations. The earlier works could have been presented in more clear manner, currently the paper gives an impression that learning joint function over (x,y) is novel which it surely isn't. It was a bit hard to follow the earlier works, they should be more clearly categorised.

The artificial datasets raise more questions than they answer. Both of them are simple univariate cases, where one would expect even simple approaches work very well. But there are suddenly 40000 datapoints for the first case (why not eg 50?), optimisation seems to take hundreds of thousands of iterations, and the results are still quite bad all around (fig3). I suspect that there are coding issues, and overall the univariate examples are not very informative in general. The "high-dimensional" case is not really high-dimensional, and a real one should have been proposed instead. There are also no real competing methods in any experiments, except for very simple regression methods or baselines from the 90's (MDN). With no comparisons to competing methods the experiments are close to worthless. The two real dataset experiments are not enough, and even there the improvement seems to be modest with only trivial baselines included. I also don't understand why the bike classification problem is chosen for regression.

The paper proposes to combine the implicit function idea with output mode estimation, however the problem definition is vague, competing Bayesian methods and density estimation methods are ignored, the experiments are insufficient with little state-of-the-art comparisons, few datasets and clear problems in the learning. The results show only small improvements, which are not explicated sufficiently. The specific problem tackled is of minor importance for the wider ICLR community.

**Experience Assessment:**

I do not know much about this area.

**Review Assessment: Checking Correctness Of Derivations And Theory:**

I assessed the sensibility of the derivations and theory.

**Review Assessment: Checking Correctness Of Experiments:**

I assessed the sensibility of the experiments.

**Review Assessment: Thoroughness In Paper Reading:**

I made a quick assessment of this paper.

---

> ### Author Response · Authors · 2019-11-15
> **Thank you for reading our paper.**
>
> Thank you for reading our paper. We believe our empirical results should be considered as persuasive in modal regression literature as far as we know.

---

### Official Review · AnonReviewer3 · 2019-10-22
**Official Blind Review #3**

**Rating:** 3

**Review:**

The paper proposes a parametric modal regression algorithm for multi-modal distributions. In such settings, learning the conditional mode is more desirable than the conditional mean, the latter being the focus of standard regression approaches. The objective in model regression is to find the conditional mode.

Existing non-parametric approaches to learning the conditional mode are difficult to scale. On the other hand, existing parametric approaches for modal regression aim to learn the full conditional distribution, which is a much harder problem. The present work proposes a parametric approach to estimate the conditional mode using the Implicit Function Theorem. The key idea is to learn a (parametric and implicit) function f(x,y) whose minima(s) 'y' corresponds to the conditional modes. However, the paper does not explain clearly how the above idea is put to practice as an algorithm in Section 3. It will be great if Section 3 can be re-written with more detailed explanations in paragraphs following the "Implicit Function Theorem" definition and more examples.

The experiments mainly are performed artificial datasets to assess the algorithm when the underlying multi-model distributions are known. Empirical results are also discussed on a couple of real-world datasets where the proposed algorithm obtains the best results.

**Experience Assessment:**

I do not know much about this area.

**Review Assessment: Checking Correctness Of Derivations And Theory:**

I assessed the sensibility of the derivations and theory.

**Review Assessment: Checking Correctness Of Experiments:**

I did not assess the experiments.

**Review Assessment: Thoroughness In Paper Reading:**

I made a quick assessment of this paper.

---

> ### Author Response · Authors · 2019-11-15
> **Thank you for reading our paper.**
>
> Thank you for taking time to read our paper. We believe our empirical results should be considered as persuasive in modal regression literature as far as we know.

---

### Official Review · AnonReviewer1 · 2019-10-22
**Official Blind Review #1**

**Rating:** 6

**Review:**

This paper considers the regression problem in scenarios in which the conditional distribution of the response variable y given the input x is multimodal. For artificially constructed datasets, in which the the conditional modes are known, performance is assessed by the RMS distance to the mode closest to each prediction.  For real-world datasets, performance is in terms of RMSE and MAE.

The authors propose to learn a function f(x,y) that takes both an input x and a proposed output y and produces a value of 0 when the output y is correct for the input (i.e a conditional mode of p(y|x)), and hopefully a nonzero output when y is incorrect.  To predict on a point x, one searches for a y that makes f(x,y)=0.  For this to work, we will need f(x,y)≠0 for incorrect y. To achieve this, the authors take inspiration from the implicit function theorem and additionally seek an f with ∂f/∂y=-1 at the modes for each x.  For implementation, they seeks an f that minimizes the objective f²(xᵢ,yᵢ) + ( ∂f(xᵢ,yᵢ)/∂y - 1 )² over the training data.  To predict y for a given x, they find argmin_y f²(x,y) + ( ∂f(x,y)/∂y - 1 )² by grid search over the range of y, which they assume to be known.

I think this is a very interesting and novel approach to regression.  It seems to work well in artificial examples and competitively on some real world examples.

My main concern is that, although the objective function is fairly intuitive, it is not  clear to me what the properties are of the population minimizer (i.e. in the infinite data case).  Equation (1) defines modes as local maxima of the density, and for evaluation purposes on the artificial data, we look at deviation from the closest conditional mode.  With infinite data, do we expect the f to converge to something that's 0 at all the conditional modes, or just at the ones with the largest density?  What do we even want to happen?  Can we use this approach to predict all the modes for each x?  While I consider all of these interesting questions, after some consideration, I don't think that missing answers to these questions should delay publication.

I recommend accepting this paper, assuming reasonable responses can be provided to the questions and issues below.  It's a novel and promising approach to an interesting problem in regression.  It leaves some open questions for further theoretical analysis, but that seems ok to me.

Specific questions:
- In section 3, second sentence, I think you want to write either the minimum of f^2 or a zero of f?
- Page 3, not sure what is meant by "should be described by the same mapping g_j(x)."
- In the implicit function theorem, it ends with a capital F, but I think you want f.
- When you say "this condition is only local and may not encourage a minimal set of conditional modes" -- I guess you mean a minimum set of zeros? Or predicted conditional modes?
- In Equation 5, you write a full l_2 norm for what I believe is just a scalar -- how about parenthesis instead?
- When you make your y prediction using the argmin... it seems like you could also minimize without the partial derivative term, as in your original explanation of f.  Did you compare the performance with and without the partial derivative term?
- In comparing to the MDN network, it's interesting that MDN needs more mixture components than the true distribution, but that doesn't really seem like a point of criticism.  Neural networks themselves seem to work better with far more parameters than we think they should need.  To be fair, I'd suggest hyperparameter searching over a broader range of mixture components for MDNs, maybe up to 10.
- In your section "verify the learned error distribution", you say that "it is obvious that the error function indeed looks like a Gaussian distribution and the one trained without noise 4(a) shows an extremely small variance."  Two comments: 1) It's definitely not obvious that 4(a) has a Gaussian distribution. 2) Why would you expect 2a to be Gaussian?  The variance should be entirely due to the randomness in the selection of training data, which has nothing to do with Gaussians in the 0 noise case, as far as I can tell.
- In Figure 5, are the error bands the standard error of the mean, or the standard deviation across trials, or something else?
- I don't at all understand the setup of the 'Examining the neural network representation'.  I understand what the embedding of x is for the L2 model but what does that mean for the Implicit model, in which an x cannot be embedded without a y?


**Experience Assessment:**

I have read many papers in this area.

**Review Assessment: Checking Correctness Of Derivations And Theory:**

N/A

**Review Assessment: Checking Correctness Of Experiments:**

I assessed the sensibility of the experiments.

**Review Assessment: Thoroughness In Paper Reading:**

I read the paper thoroughly.

---

> ### Author Response · Authors · 2019-11-15
> **Thanks for the positive feedback.**
>
> Thanks for reading our paper carefully.  We will take into account all of your suggestions/questions/comments in the next version. We apologize that we do not have sufficient time to respond to all of your questions.
>
> When doing prediction: yes, we conducted experiments with/without the partial derivative term. The partial derivative term is critical to ensure implicit function theorem applies around the query point. Empirically, doing prediction without that term can work on some dataset, but can also find many wrong points on some other datasets. Hence, incorporating that term is more robust across datasets.
>
> The error distribution is assumed to be Gaussian. "look like Gaussian" may be subjective. (a) is using a training set without adding noise so it can be thought of as Gaussian with mean zero and take the limit of variance to zero. Our figure shows that the errors highly concentrate around zero. Note that we use the whole training set for computing the distribution. It should be a good approximation.
>
> Fig 5. For implicit model, we indeed get the representation by feeding into both x and y and y is found by grid search. Implicit model takes both x and y as input. Hence we expect the neural network representation should be finer as the target space information can be utilized.

---

### Decision · Program_Chairs · 2019-12-19

**Decision:**

Reject

**Comment:**

The paper proposes an implicit function approach to learning the modes of multimodal regression. The basic idea is interesting, and is clearly related to density estimation, which the paper does not discuss.

Based on the reviews and the fact that the authors did not submit a helpful rebuttal, I recommend rejection.